# Semi-Supervised Speech Enhancement with Gradient-Guided Channel Attenuation

## Abstract

Recent methods for speech enhancement (SE) have generally adopted the supervised learning way and trained the models on synthetic noisy-clean paired speech data. However, when applying the supervised trained SE model to the recordings of real-world scenario, which we call unlabeled data, it will lead to the performance degradation. To improve the generalization performance of SE, we propose a semi-supervised monaural speech enhancement network, SS-SENet, which adopts the mean-teacher (MT) framework with domain adversarial (DA) learning to effectively exploit the unlabeled data. We also propose the Gradient-Guided Channel Attenuation (GGCA) module for suppressing the domain-specific features and enhance domain-invariant one, and Domain Shift-Aware Monitor (DSAM) strategy for dynamically adjusting the attenuation rate in GGCA. Comparing with seven SOTA methods exploiting the unlabeled data, our proposed SS-SENet achieves the best performances at all metrics both on synthetic Reverberant LibriCHiME-5 and LibriMix datasets, and at the critical metric, OVRL, on the real-world CHiME-5 dataset. The results verify that our proposed basic MT-based method is superior to the compared methods based on full supervised or self-supervised learning. It also verifies the effectiveness of our proposed GGCA module and DSAM strategy. The source code is available at
`https://anonymous.4open.science/r/SS-SENet`.

## 1 INTRODUCTION

Speech enhancement (SE) is dedicated to extracting the clean speech from mixtures under the complex acoustic environments, while suppressing the noise, reverberation and other interferences thus improving the quality and intelligibility of extracting speech (Wei et al., 2023). Current mainstream SE methods generally adopt the supervised learning way and trained the model on synthetic data. However, there still exsit the discrepancy of a certain extent in terms of acoustic condition between the synthetic and real-world recordings (Li Li, 2024). The noise and interference both have a great variety of types and intensities and are existed in real-world environment randomly and dynamically, which is differ from the assumption for creating the synthetic data (Xu et al., 2025). Besides, under the scene of crosstalking in real-world environments, it is difficult to obtain the specific pure speech, which makes the construction of labeled data impractical (Ito et al., 2023). Therefore, applying the supervised SE model trained on the synthetic datasets to the scenario of real-world will lead to performance degradation, due to the unavoidable acoustic mismatch between the synthetic and real-world recordings (Wang et al., 2024; Yao et al., 2025). Conversely, we can easily acquire a large number of noisy mixtures with various types of noise and interference in real-world scenarios, but without the corresponding clean reference signals, which are termed as unlabeled data.

Recently, leveraging the unlabeled real-world recordings to improve the performance of SE models under the condition of real-world has become a central challenge (Zhang et al., 2021; Frenkel et al., 2023; Lee et al., 2024; Frenkel et al., 2024). Frenkel et al. proposed to adpot domain adversarial (DA) learning to exploit real-world unlabeled data, enabling the model to learn the domain-invariant representations (Frenkel et al., 2024). Besides, self-supervised learning based methods , such as RemixIT (Tzinis et al., 2022a;b) and its variants (Chen et al., 2023; Fujimura et al., 2023; Li Li, 2024; Liao et al., 2025) have been proposed, and they typically adopt a two-stage pipeline, as shown in Fig. 1 (a). At the first stage, the SE model is pre-trained on synthetic dataset, which includes the noisy mixture and its corresponding clean speech and noise references. At the second stage, both the

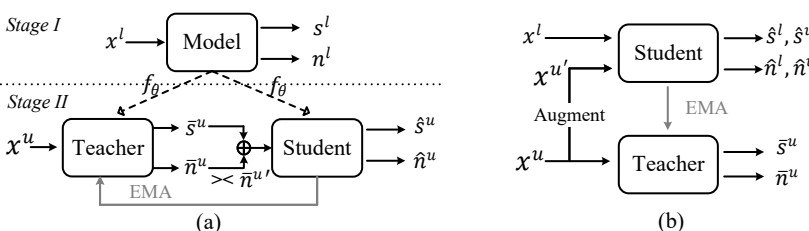

Figure 1: Illustrations of (a) self-supervised learning (RemixIT) and (b) semi-supervised learning based on MT framwork (ours). The superscripts $l$ and $u$ denote the labeled and unlabeled data, respectively. $x$ denotes the mixture, and $s$ and $n$ the speech and noise signals. $\hat{}$ and $\bar{}$ denote the signals estimated by student and teacher models, respectively. $f_\theta$ denotes the pre-trained parameter at the stage I, and $><$ the permutation operation.

teacher and student models are initialized with the pre-trained weights $f_\theta$. Only the real-world unlabeled recordings are fed into the teacher model, and the noise prediction of which are then randomly permuted and remixed with its speech prediction to generate the bootstrapped mixtures. These mixtures are used to train the student model together with the teacher's corresponding predictions, which are named as pseudo-labels (Tzinis et al., 2022a;b). Li et al. proposed the Remixed2Remixed adopting a Noise2Noise learning strategy, where two bootstrapped mixtures generated by teacher model are used to train the student model through optimizing an Noise2Noise-based cost function (Li Li, 2024). Liao et al. proposed the PHA-RemixIT which enhances the remixing diversity by using noise in labeled data, and adopts the heterogeneous noise-invariant loss and adaptive focal weight to improve the generalization across various acoustic conditions (Liao et al., 2025).

Although the aforementioned methods have begun to exploit the recordings of the real-world scenarios, the quality of estimated speech heavily relies on the fixed model pre-trained at the first stage (Wang et al., 2024; Liao et al., 2025). Besides, this two-stage training framework is cumbersome and inefficient, and completely detach from the supervised by labeled data at the second stage, which may cause the model's deviation and ultimately resulting in performance reduction (Liu et al., 2024). The semi-supervised learning based on the Mean-Teacher (MT) framework was firstly proposed by Tarvainen et al. (Tarvainen et al., 2017), which offers an end-to-end way and requires only a single parallel pass through the labeled and unlabeled data for training, as shown in Fig .1 (b). However, the semi-supervised learning based on MT framework has been mostly applied to computer vision tasks, such as image classification and object detection (Döbler et al., 2023; Huang et al., 2023; Liu et al., 2024; Qiao et al., 2024; Kumar et al., 2025), but to our knowledge, it is underexplored for the regression-based tasks like speech separation.

In this paper, we propose a novel **S**emi-**S**upervised monaural **S**peech **E**nhancement **N**etwork, SS-SENet, which is the first attempt to apply the semi-supervised learning of MT-based for SE task. As shown in Fig. 2, SS-SENet adopts a MT structure and introduces a domain discriminator with gradient reversal layer (GRL) to minimize the gap of feature distributions between synthetic and real-world data (Ganin et al., 2016; Frenkel et al., 2024). We also propose the Gradient-Guided Channel Attenuation (GGCA) to selectively attenuate the features carrying strong domain-specific information, and the Domain Shift-Aware Monitor (DSAM) to adjust the attenuation rate in GGCA by monitoring the extent of domain-shift.

We trained the SS-SENet on both synthetic labeled dataset, LibriMix (Cosentino et al., 2020) and real-world unlabeled dataset, CHiME-5 (Barker et al., 2018), and evaluated it across three datasets including LibriMix, CHiME-5, and Reveberant LibriCHiME-5. In summary, our contributions are as follows:

**1)** We make the first attempt to introduce the semi-supervised learning of MT-based for SE task, and propose SS-SENet, which can effectively expoit the real-world unlabeled data for training and improve the adaptation of SE model under the real-world scenarios.

**2)** We propose Gradient-Guide Channel Attenuation (GGCA) for attenuating the domain-specific features thus extracting the domain-invariant oneD, and adopt domain adversarial (DA) learning for reducing the feature-level domain discrepancy.

**3)** We propose Domain Shift-Aware Monitor (DSAM) for adjusting the attenuation rate in GGCA by real-time monitoring the extent of domain-shit between the synthetic and real-world data.

## 2 RELATED WORKS

**SE Based on Supervised Learning.** SE methods based on deep neural networks have achieved remarkable success in supervised settings, as they benefit from large-scale synthetic noisy-clean paired speech data. Existing methods can be broadly categorized into time-domain (T-domain) (Wang et al., 2021; Zhao & Ma, 2023) and time-frequency domain (TF-domain) based methods (Lu et al., 2023; Chao et al., 2024; Abdulatif et al., 2024; Yan et al., 2025; Hu et al., 2025). These methods typically rely on the assumption that training and testing data are from similar distributions (Liao et al., 2025).

**SE Based on Self-Supervised Learning.** Such supervised SE models would result in performance degradation of a certain degree when inferring on the real-world dataset. Recent SE researches have begun to focus on the exploiting of unlabeled data. DA learning is adopted to enable the model to learn the domain-invariant representations between the synthetic and real-world recordings (Frenkel et al., 2023; 2024; Lee et al., 2024). RemixIT and its variants based on self-supervised learning, adopt a two-stage pipeline, where the SE model is pre-trained on the synthetic data, and the pre-tarined teacher model generates the pseudo-label of unlabeled date for training the student model (Tzinis et al., 2022a;b; Chen et al., 2023; Fujimura et al., 2023; Li Li, 2024; Liao et al., 2025).

**Semi-supervised Learning Based on MT Framework.** MT framework is widely used in semi-supervised learning that maintains a student-teacher model pair with identical architectures (Tarvainen et al., 2017). Its core idea is to enforce consistency between the predictions of student and teacher models on the unlabeled data (Cai et al., 2019). To reduce the influence of unreliable pseudo-labels in early training, the consistency loss is typically weighted by a time-dependent schedule (French et al., 2018). The semi-supervised learning based on MT framework has been extensively studied in computer vision tasks (Döbler et al., 2023; Huang et al., 2023; Liu et al., 2024; Qiao et al., 2024; Kumar et al., 2025), but has not been explored for regression-based SE task.

## 3 METHOD

Our proposed SS-SENet adopts the semi-supervised learning of MT-based and DA training strategy. Let $\mathcal{D}^l = \{(x_i^l, s_i^l, n_i^l)\}_{i=1}^{N^l}$ denotes the labeled dataset, where $x^l$, $s^l$ and $n^l$ denote the noisy mixture, clean speech and pure noise reference. $\mathcal{D}^u = \{x_i^u\}_{i=1}^{N^u}$ denotes unlabeled real-world data, where $i$ is the index , and $N^l$ and $N^u$ the total number of samples for labeled and unlabeled datasets. Three objective functions are used for training the student model, which includes the supervised loss $\mathcal{L}_{\text{Sup.}}$ based on $\mathcal{D}^l$, consistency loss $\mathcal{L}_{\text{Cons.}}$ based on $\mathcal{D}^u$, and domain discrimination loss $\mathcal{L}_{\text{Dom.}}$ based on both $\mathcal{D}^l$ and $\mathcal{D}^u$, as shown in Fig. 2. The first two losses are called semi-supervised learning loss while the last one DA training loss.

### 3.1 MEAN-TEACHER FRAMEWORK FOR SEMI-SUPERVISED SPEECH ENHANCEMENT

SS-SENet consists of a teacher–student model pair adopting the same SE backbone, where the student model differs from teacher one in that there is a GGCA module including DSAM strategy between the encoder and bottleneck. During the training phase, the teacher model only processes the original unlabeled samples $x^u$ while the student one processes the labeled samples $x^l$ and augmented version of unlabeled samples $x^u$. The augmented unlabelled samples $x^{u'}$ are generated by adopting a remixing-based strategy (Tzinis et al., 2022a;b), by which it can constructs new mixtures by permuting the teacher's predicted noise and remixing them with the teacher's predicted speech.

For $\mathcal{D}^l$, the student model is trained in a fully-supervised manner by aligning its predictions with the ground-truth, including clean speech $s^l$ and noise reference $n^l$. For $\mathcal{D}^u$, the predictions of teacher

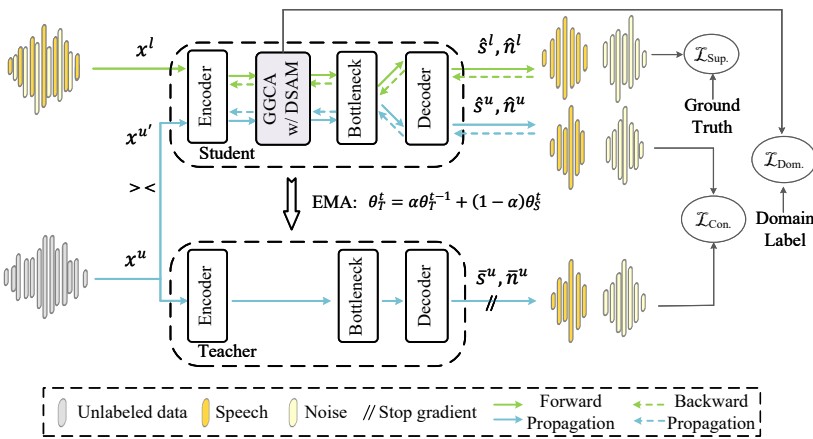

Figure 2: Overview of SS-SENet. // denotes stop grandient. EMA means exponential moving average. $\theta_T$ and $\theta_S$ denote the parameters of teacher and student models, respectively. $t$ is the training step, and $\alpha \in [0, 1]$ the decay rate. $\mathcal{L}_{\text{Sup.}}$, $\mathcal{L}_{\text{Cons.}}$ and $\mathcal{L}_{\text{Dom.}}$ denote the supervised, consistency and domain discrimination losses, respectively.

model are servered as the pseudo-labels for training the student model, which is encouraged to produce the consistent predictions with that by the teacher model. Through this joint optimization, the student model is updated simultaneously based on accurate supervision on the labeled data and regularization on the unlabeled data, while the teacher model is updated via exponential moving average (EMA) of student's parameters, $\theta_T^{(t)} = \alpha \theta_T^{(t-1)} + (1-\alpha)\theta_S^{(t)}$, and its pseudo-label quality is progressively improved. Where the $\theta_T$ and $\theta_S$ are the parameters of teacher and student models, respectively. $t$ denote the training step, and $\alpha \in [0, 1]$ the decay rate.

### 3.2 GRADIENT-GUIDE CHANNEL ATTENUATION (GGCA)

During DA training, the feature of each channel are treated equally, ignoring the fact that domain-specific information may be concentrated on a part of channels not all. Inspired by the research on domain generalization (Guo et al., 2023; Hui et al., 2024), we propose GGCA to selectively attenuate the domain-specific features and enhance domain-invariant one according to the gradients of domain discriminator.

As shown in Fig. 3, given the feature maps $\mathbf{F}_E(x^*)$ extracted by the encoder, where $* \in u, l$, we use a global average pooling (GAP) layer to obtain feature vectors $P^*$, which are then fed into a domain discriminator (D) for domain discrimination. To avoid the negative impact of domain discriminator on the main network, we use a gradient reversal layer (GRL) before GAP to reverse the gradients of domain discrimination loss, $\mathcal{L}_{\text{Dom.}}$, which is defined by:

$$\mathcal{L}_{\text{Dom.}} = -d \log D(P_i) - (1-d) \log(1 - D(P_i)), \tag{1}$$

where $d$ is the domain label. The samples from labelled dataset are labeled as $d=0$, and that from unlabelled one, $d=1$.

We exploit the gradient values of domain discriminator to determine that the features of which channels may contain more domain-specific information. Generally, the features contribute the most to the prediction of domain are likely to contain the most domain-specific information. To measure how much the features of each channel contribute to domain prediction, we define a domain discriminability metric of feature $F_E(x^*)$ with the channels number $C$ as:

$$w_c^* = \frac{\partial \mathcal{L}_{\text{Dom.}}}{\partial F_E(x^*)}, \quad c \in \{1, 2, ..., C\}, \tag{2}$$

where $C$ is the channels number. The larger value $w_c^*$ is, the more domain-specific information the feature of $c$-th channel contains. Therefore, the features of those channels should be attenuated

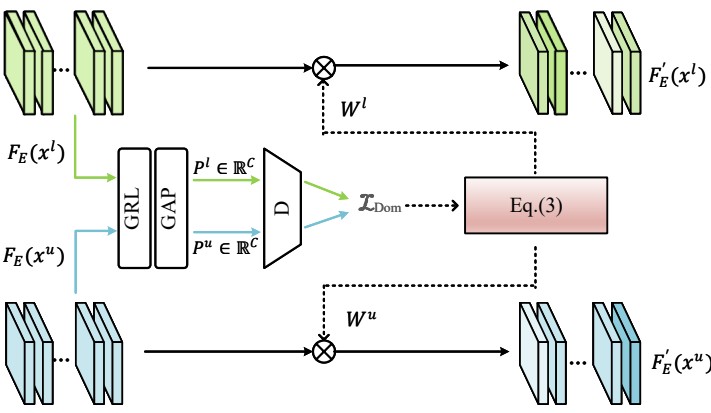

Figure 3: Sturcture of the GGCA. $\mathbf{F}_E(\cdot)$ denotes the output of encoder, and $W$ the attenuation matrix of $\mathbf{F}_E(\cdot)$. $\otimes$ dnotes element-wise multiplication. $\mathbf{F}'_E(\cdot)$ denotes the final feature representations.

to alleviate the problem of domain-shift, which also denotes the discrepancy between the features extratced from synthetic and real-world data. We rank the values of $w_c^*$ and select the corresponding features of K channels with $w_c^*$ value of the largest Top-k:

$$W^* = \begin{cases} \sigma(-w_c^*), & \text{if } c \in \text{Top}(w_c^*, K) \text{ and } w_c^* > 0 \\ 1, & \text{otherwise} \end{cases} \tag{3}$$

Here, $\sigma(\cdot)$ denotes the sigmoid function, $W^*$ the final attenuation matrix applied for the $\mathbf{F}_E(x^*)$. The number of selected channels $K = C \cdot r$, where $r$ is the attenuation ratio and its calculation is referred to 3.3. The finally feature is obtained: $F'_E(x^*) = W^* \cdot F_E(x^*)$.

### 3.3 DOMAIN SHIFT-AWARE MONITOR (DSAM)

The discrepancy in acoustic conditions between the labeled and unlabeled data may lead to domain shift, the extent of which fluctuates dynamically during training. Adpoting a fixed attenuation ratio $r$ in GGCA may cause suboptimal performance (Feng et al., 2025).

Consequently, we propose the DSAM strategy for obtaining adaptive attenuation ratio $r$, which can monitor the extent of domain shift by measuring the variance of intermediate features extracted from labeled and unlabeled data during the training. Specifically, we update the attenuation ratio $r$ in GGCA by average channel-wise variance $\nu$ via a Sigmoid function:

$$r = \epsilon \cdot \frac{1}{1 + \exp(-k(\nu - s))} \tag{4}$$

where $\epsilon \in (0, 1)$ is a scaling factor that controlling the maximum attenuation rate. $k$ and $s$ are parameters of the Sigmoid function that determine its slope and offset, respectively. The average channel-wise variance $\nu$ is defined as follows:

$$\nu = \frac{1}{C} \sum_{c=1}^{C} \left( \frac{1}{n^l + n^u - 1} \left( \sum_{i=1}^{n^l} (P_{i,c}^l - \bar{P}_c)^2 + \sum_{j=1}^{n^u} (P_{j,c}^u - \bar{P}_c)^2 \right) \right) \tag{5}$$

where $n^l$ and $n^u$ are the number of labeled and unlabeled samples in a mini-batch. $\bar{P}_c$ is the mean value of features of channel $c$ across both labeled and unlabeled data, which is defined as follows:

$$\bar{P}_c = \frac{1}{n^l + n^u} \left( \sum_{i=1}^{n^l} P_{i,c}^l + \sum_{j=1}^{n^u} P_{j,c}^u \right) \tag{6}$$

A larger value of $\nu$ suggests a more severe domain mismatch, requiring more aggressive attenuation of domain-specific features, while a smaller value of $\nu$ indicates a slighter domain mismatch, and

Table 1: Comparison with seven SOTA methods exploiting unlabeled data on three datasets: real-world CHiME-5 dataset, and synthetic LibriMix and Reverberant LibriCHiME-5 datasets, the last of which was excluded from training. The SS-SENet adopting the backbones of SuDoRM-RF (T-domain) and BSRNN (TF-domain) are evaluated for fair comparison. Within each group based on T- or TF-domain, **bold** and underline denote best and second-best performance. The 23'CHiME refers to the official outcomes published for the Task2 of CHiME 2023 (CHiME-7 UDASE). The ⋆ denotes that we implemented based on the code supplied officially (Tzinis et al., 2022b).

| Method | Backbone | CHiME-5 (Real-World) | | | Rever. LibriCHiME-5 (Synthetic) | | LibriMix (Synthetic) | | PUB. |
|---|---|---|---|---|---|---|---|---|---|
| | | SIG | BAK | OVRL | SI-SDR | PESQ | SI-SDR | PESQ | |
| noisy | - | 3.48 | 2.92 | 2.84 | 6.59 | 1.55 | 4.91 | 1.25 | - |
| | | | | T-domain | | | | | |
| RemixIT⋆ | SuDoRM-RF | 3.26 | 3.64 | 2.82 | 9.44 | 1.68 | 11.47 | 1.87 | 22'JSTSP |
| Sogang ISDS1 | SuDoRM-RF | **3.39** | 3.60 | 2.90 | 12.42 | - | - | - | 23'CHiME |
| Sogang ISDS2 | MossFormer | 3.32 | 3.70 | 2.88 | 12.42 | - | - | - | 23'CHiME |
| Remixed2Remixed | SuDoRM-RF | 3.35 | 3.42 | 2.85 | 12.41 | - | - | - | 24'ICASSP |
| **SS-SENet (Ours)** | SuDoRM-RF | 3.34 | **3.85** | **2.97** | **13.60** | **1.89** | **13.04** | **2.16** | |
| | | | | TF-domain | | | | | |
| Remixit-G | Uformer | 3.39 | 3.93 | 3.07 | 12.95 | - | 8.83 | - | 23'CHiME |
| Multi-CMGAN+/+ | CMGAN | 3.49 | 3.86 | 3.12 | 6.95 | - | - | - | 24'ICASSP |
| PHA-RemixIT | BSRNN | 3.46 | **4.03** | 3.22 | 11.6 | - | - | - | 25'ICASSP |
| **SS-SENet (Ours)** | BSRNN | **3.54** | 3.87 | **3.30** | **13.00** | **1.89** | **12.16** | **1.87** | |

thus requiring more gentle attenuation. Inspired by the learnable sigmoid function, where the slope is a learnable parameter (Fu et al., 2021; Hu et al., 2025), we propose a dynamic adjustment strategy where the value of $k$ in Eq. 5 is adjusted in real-time according to $\nu$, $k(\nu) = \gamma \cdot \nu$, where $\gamma$ serves as a scaling factor and set to 20.

## 3.4 LOSS FUNCTION

The student model is optimized using three loss functions: the supervised loss $\mathcal{L}_{\text{Sup.}}$, consistency loss $\mathcal{L}_{\text{Cons.}}$, and domain discrimination loss $\mathcal{L}_{\text{Dom.}}$. The total loss $\mathcal{L}_{\text{Total}}$ is defined as follows:

$$\mathcal{L}_{\text{Total}} = \mathcal{L}_{\text{Sup.}} + \lambda_1 \cdot w(t)\mathcal{L}_{\text{Cons.}} + \lambda_2 \cdot \mathcal{L}_{\text{Dom.}}, \tag{7}$$

where $\lambda_1$ and $\lambda_2$ are hyperparameters. $w(t)$ is a ramp-up function for dynamically adjusting the weight of consistency loss: $w(t) = \exp\left[-5\left(1 - \frac{t^2}{\mathcal{T}}\right)\right]$ where $t$ denotes the current iteration of training, and $\mathcal{T}$ the ramp-up length and set to the multiplication of the iterations number for each epoch and 25 (epoch), in our implementation. The $\mathcal{L}_{\text{Sup}}$ is based on the labeled data $\mathcal{D}^l$, which encourages the SE model to accurately reconstruct both the clean speech and noise signals.

$$\mathcal{L}_{\text{Sup.}} = \mathcal{L}_{\text{rec.}}(s^l, \hat{s}^l) + \mathcal{L}_{\text{rec.}}(n^l, \hat{n}^l), \tag{8}$$

where the $\mathcal{L}_{\text{rec.}}$ denotes any desired signal-level reconstruction loss function, which is a weighted sum of the mean square error (MSE) and negative scale-invariant signal-to-distortion ratio (SI-SDR) losses, with corresponding weights of 1.0 and 0.5, respectively. $\hat{s}^l, \hat{n}^l$ denote the predicted speech and noise of the student model, and $s^l, n^l$ the corresponding reference signals.

The $\mathcal{L}_{\text{Cons.}}$ enforces consistency between the predictions of student and teacher models on the unlabeled data $\mathcal{D}^u$:

$$\mathcal{L}_{\text{Cons.}} = \mathcal{L}_{\text{rec.}}(\bar{s}^u, \hat{s}^u) + \mathcal{L}_{\text{rec.}}(\bar{n}^u, \hat{n}^u), \tag{9}$$

where $\bar{s}^u, \bar{n}^u$ are the predicted speech and noise of teacher model, and $\hat{s}^u, \hat{n}^u$ are that of the student model. The domain discrimination loss $\mathcal{L}_{\text{Dom.}}$ is computed as in Eq. 1.

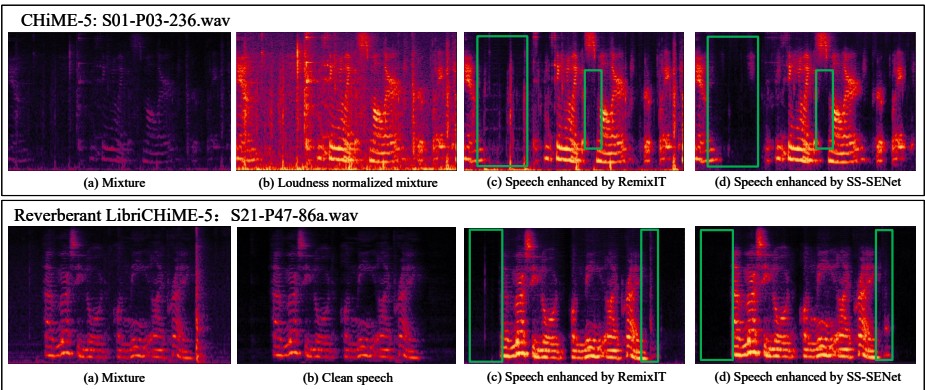

Figure 4: Visualization analysis of spectrograms of two recordings from two datasets. Subfigures of each column from (a)–(d) denote the spectrograms of noisy mixture, clean speech (or loudness-normalized mixture for the unlabeled CHiME-5 dataset), and the estimated speech by RemixIT and SS-SENet, respectively. The highlighted boxes indicate the regions with obvious improvement achieved by SS-SENet compared with that by RemixIT.

## 4 EXPERIMENTAL SETUP

**Dataset.** We conducted experiments on three datasets: the CHiME-5 only with unlabeled data (Barker et al., 2018), LibriMix (Cosentino et al., 2020) and Reverberant LibriCHiME-5 both with labeled data. Where the first two datasets are used for training, evaluating and testing while the latest one for evaluating and testing. For the details of three datasets, please refer to Appendix A.

**Training and evaluation.** We trained the models for 100 epochs on three NVIDIA A40 GPUs using the Adam optimizer with an initial learning rate of 0.001 and batch size of 42. Each mini-batch contains the same number of labeled and unlabeled samples. The decay rate $\alpha$ in EMA is set to 0.99. After the ramp-up strategy is finished, the learning rate will be reduced by a factor of 3 at every 15 epochs. For the $eval/1$ subset of CHiME-5 dataset, only including real-world unlabeled recordings, we calculated the DNSMOS (Reddy et al., 2022) scores for speech (SIG), background noise (BAK), and overall (OVRL) qualities. For the complete evaluation set of both Reverberant LibriCHiME-5 and LibriMix datasets, only including synthetic labeled recordings, we calculated the SI-SDR and perceptual evaluation of speech quality (PESQ). For all metrics, higher scores indicate better performance. Further metric details are provided in Appendix B.

## 5 RESULTS

### 5.1 COMPARISON WITH STATE-OF-THE-ART METHODS

We compare the SS-SENet with seven state-of-the-art (SOTA) methods on three datasets, each of them exploit the real-world unlabeled data, including RemixIT (Tzinis et al., 2022b), Remixed2Remixed (Li Li, 2024), PHA-RemixIT (Liao et al., 2025), Mutil-CMGAN+/+ (Close et al., 2024), as well as Sogang ISDS1/2 (Jang et al., 2023) and RemixIT-G (Zhang et al., 2023). Where the last three methods are pubulished for the Task2 of CHiME 2023 (CHiME-7 UDASE) [1]. These compared methods include four T-domain-based and three TF-domain-based ones. For fair comparison, SS-SENet was evaluated using the SuDoRM-RF (T-domain) (Tzinis et al., 2020) and BSRNN (TF-domain) (Yu et al., 2023) backbones, and the corresponding results are listed in Table 1. For two groups of T- and TF-domain-based methods, SS-SENet achieves best performance on both Reverberant LibriCHiME-5 and LibriMix datasets, which consist entirely of synthetic labeled data. For CHiME-5 dataset consisting of unlabeled data, SS-SENet also achieves the best performance among the compared methods of two groups on the score of OVRL, which is a critical metric.

---

[1]https://www.chimechallenge.org/challenges/chime7/task2/results

Table 2: Comparison of the fully **supervised** SuDoRM-RF (F), **self-supervised** RemixIT and **semi-supervised** MT-based methods on three datasets. The CHiME-5 dataset consists of unlabeled real-world recordings. All methods adopt the backbone of SuDoRM-RF to ensure a fair comparison.

| Method | Training Data | | CHiME-5 (Real-World Data) | | | Rever. LibriCHiME-5 (Synthetic Data) | | LibriMix (Synthetic Data) | |
|---|---|---|---|---|---|---|---|---|---|
| | LibriMix | CHiME-5 | SIG | BAK | OVRL | SI-SDR | PESQ | SI-SDR | PESQ |
| Noisy | - | - | **3.478** | 2.917 | 2.839 | 6.589 | 1.547 | 4.909 | 1.245 |
| SuDoRM-RF (F) | ✓ | ✕ | 3.330 | 3.590 | 2.879 | 7.805 | 1.566 | **13.235** | **2.195** |
| RemixIT | ✓ | ✓ | 3.255 | 3.644 | 2.824 | 9.440 | 1.678 | 11.470 | 1.869 |
| **Basic MT (Ours)** | ✓ | ✓ | 3.333 | **3.761** | **2.930** | **13.460** | **1.930** | 13.160 | 2.192 |

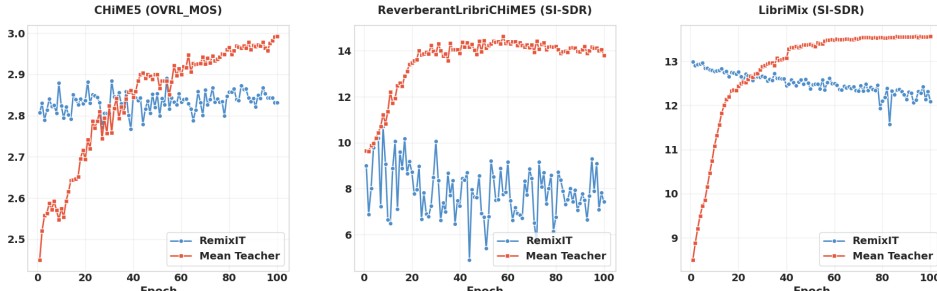

Figure 5: Evaluation performance trends of RemixIT (blue curve) and our proposed basic mean-teacher (MT) framework (red curve) on the development sets of three datasets. The vertical axis of the leftmost figure shows the scores of OVRL on the CHiME-5 dataset, and that of the middle and rightmost one that both of SI-SDR on Reverberant LibriCHiME-5 and LibriMix datasets, respectively. The horizontal axis denotes the epoch number.

We also draw visualization analysis of spectrograms of two recordings from two datasets. As shown in Fig. 4, the subfigures of each column from (a)–(d) denote the spectrograms of noisy mixture, clean speech (or loudness-normalized mixture) and the speech predicted by RemixIT and SS-SENet, respectively. For the unlabeled recording in CHiME-5 dataset, subfigure (b) is the spectrogram of the mixture whose loudness has been normalized to -30 LUFS (Loudness Unit Full Scale) using the Python package $pyloudnorm$ (Steinmetz & Reiss, 2021) (The reason for loudness normalizing is explained in the Appendix B). The highlighted boxes indicate the regions where SS-SENet achieves obvious improvement compared to RemixIT. Comparing the subfigures (c) with (d) of the first row in Fig. 4, it is obviously observed that some non-speech components still remain in the areas highlighted by green boxes. As for demos of SE, please refer to the project page [2].

## 5.2 ABLATION STUDY

We conducted a series of ablation experiments on three datasets to verify the effectiveness of the semi-supervised learning of MT-based and three key components in SS-SENet.

**Semi-supervised learning of MT-based framework.** To evaluate the effectiveness of the semi-supervised learning of MT-based framework, we compare it with the fully supervised SuDoRM-RF (F) and self-supervised RemixIT on three datasets. The basic MT means that our proposed semi-supervised method is just based on MT framework but without three key components including DA training, GGCA and DSAM. During the training phase, SuDoRM-RF (F) was trained only on the LibriMix dataset, while the RemixIT and basic MT both on the LibriMix and CHiME-5 datasets. All methods adopted the same backbone of SuDoRM-RF (Tzinis et al., 2020) to ensure a fair comparison. As shown in Table 2, our proposed basic MT framework achieves the best performance

---

[2]https://sssenet.github.io/Demo/

Table 3: Ablation studies of three key components in SS-SENet on three evaluation sets. A1 denotes the semi-supervised learning of MT-based with DA training strategy, and A2 that with DA and GGCA. SS-SENet is the complete model including three key components.

| Mtethod | | CHiME-5 (Real-World Data) | | | Rever. LibriCHiME-5 (Synthetic Data) | | LibriMix (Synthetic Data) | |
|---|---|---|---|---|---|---|---|---|
| | | SIG | BAK | OVRL | SI-SDR | PESQ | SI-SDR | PESQ |
| A0 | MT | 3.333 | 3.761 | 2.930 | 13.460 | **1.930** | **13.160** | **2.192** |
| A1 | +DA | 3.341 | 3.821 | 2.958 | **13.612** | 1.905 | 13.024 | 2.156 |
| A2 | +GGCA | **3.352** | 3.832 | 2.964 | 13.497 | 1.888 | 13.055 | 2.170 |
| SS-SENet | +DSAM | 3.338 | **3.848** | **2.971** | 13.601 | 1.894 | 13.043 | 2.164 |

on the Reverberant LibriCHiME-5 and CHiME-5 datasets, except the score of SIG, and the second-best performance on the LibriMix dataset, while the fully supervised SuDoRM-RF (F) trained on LibriMix dataset achieves best performance on it.

We also plotted the trends of evaluation performances of the RemixIT and our proposed basic MT method, both of which were evaluated on the development sets of three datasets. As shown in Fig.5, as the number of epochs increases, the basic MT method achieves the growing performances on three datasets, while RemixIT exhibits a fluctuating and even stagnant one.

**Three key components.** Futhermore, building upon the basic semi-supervised SE of MT-based framework, we conducted a series of ablation experiments to verify the effectiveness of three components of SS-SENet, including DA learning, GGCA module, and DSAM strategy. In Table 3, A1 denotes the basic MT framework with DA learning, A2 that with DA learning and GGCA, SS-SENet the complete model including basic MT, DA, GGCA and DSAM. Note that A2 corresponds to SS-SENet without the DSAM component, which means that the attenuation ratio $r$ in GGCA is fixed rather than dynamically adjusted. In this setup, the value of $r$ was fixed at 0.1

It is clear that with DA training, the model achieves better performance on real-world CHiME-5 dataset. Introducing GGCA, the model achieves substantial improvements on all metrics for the CHiME-5 dataset. These results indicate that the features of channel-level attenuation based on the $\mathcal{L}_{\text{Dom.}}$ can help to reduce the speech distortion and background noise, thereby improving the overall speech quality. However, we observe a slight performance decrease in SI-SDR and PESQ scores on the Reverberant LibriCHiME-5 dataset. This suggests that although GGCA can improve overall performance, its use of a fixed attenuation ratio fails to adapt to the dynamically varying degree of domain-shift, thus resulting in suboptimal performance under complex acoustic conditions (To visualize the effect of GGCA, we present a comparison of spectrograms with and without the GGCA module in Appendix C). When introducing the DSAM strategy, the model achieves better performance. For CHiME-5 dataset, the scores of BAK and OVRL are further improved from 3.832 to 3.848 and from 2.964 to 2.971, respectively, while for Reverberant LibriCHiME-5, the SI-SDR and PESQ scores recover from 13.497 to 13.601 and from 1.888 to 1.894. These results verify that using DSAM for adaptively adjusting the attenuation ratio $r$ is better than using a fixed one. Additional ablation studies of maximum attenuation ratio $\epsilon$ in Eq. 4, the adjustment coefficient $k$ in DSAM, and consistency loss weight $\lambda_1$ are provided in Appendix D for completeness.

## 6 CONCLUSION

In this paper, we propose SS-SENet, which is the first attempt to apply the semi-supervised learning method of MT-based for SE task. We propose the Gradient-Guided Channel Attenuation (GGCA) module for selectively attenuating the channel-level features, and the Domain Shift-Aware Monitor (DSAM) strategy for dynamically adjusting the extent of attenuation within GGCA module. Compared with seven state-of-the-art methods that all exploit unlabeled data, our proposed SS-SENet achieves the best performances across all metrics on the Reverberant LibriCHiME-5 and LibriMix datasets, and on the critical OVRL metric for CHiME-5 dataset. The ablation results verify that our proposed basic model adopting semi-supervised learning method of MT-based is superior to the compared methods, which adopt the fully supervised or self-supervised learning methods. They also demonstrate the effectiveness of our proposed GGCA module and DSAM strategy.

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

# A DATASET DETALS

**CHiME-5:** Real multi-speaker conversational recordings between multiple speakers from 20 dinner parties in different homes with three recording locations per home (kitchen, dining room, living room) (Barker et al., 2018), captured by binaural microphones in noisy and reverberant environments. Following the CHiME-7 protocol (Leglaive et al., 2023), only the right channel is used and discarded unreliable portions of the recordings. The extracted audio segments contain up to three simultaneously-active reverberant speakers and background noise. The noisy speech signals are not labeled with the clean speech reference signals. According to (Leglaive et al., 2023), the training set consists of raw single-channel audio segments extracted from the binaural recordings. For the development and evaluation sets, segments were extracted in successive stages: (i) segments containing only background noise; (ii) remaining segments with a single active speaker and no overlap; (iii) remaining segments with up to two simultaneous speakers; and (iv) remaining segments with up to three simultaneous speakers. Noise-only segments are used to create the Reverberant LibriCHiME-5 dataset for objective evaluations. Other subsets are further divided for train (≈83h), development (≈15.5h), and evaluation (≈7h), respectively.

**LibriMix:** This dataset was originally developed for speech separation in noisy environments, it is derived from LibriSpeech clean utterances (Panayotov et al., 2015) and WHAM! noises (Wichern et al., 2019). The Libri2Mix and Libri3Mix versions of the dataset contain noisy speech mixtures with 2 and 3 overlapping speakers, respectively. A singlespeaker version of LibriMix (Libri1Mix) can be obtained by simply discarding one of the two speakers in Libri2Mix mixtures.

**Reverberant LibriCHiME-5:** In real-world conditions, particularly for the CHiME-5 recordings, it is impossible to have access to the ground-truth clean speech reference signals associated with the noisy speech mixtures. Yet, when developing and evaluating a speech enhancement algorithm it is necessary to compute objective performance metrics. For this purpose, the reverberant LibriCHiME-5 dataset is created for development and evaluation only. This dataset consist of reverberant speech and noise, with up to three simultaneously active speakers, labeled with the clean reference speech signals. Noise signals were extracted from the CHiME-5 recordings, and clean speech utterances were taken form the LibriSpeech dataset and were convolved with room impulse responses (RIRs) from VioceHome Corps (Bertin et al., 2016). The mixtures are generated by adding noise segments into randomly sampled speech utterances convolved with randomly sampled RIRs, where the SNR for each speaker is distributed as a Gaussian with a mean of 5 dB and a standard deviation (std) of 7 dB to match the CHiME-5 dataset. The proportion of 1-spk, 2-spk, and 3-spk subsets was 0.6, 0.35, and 0.05, respectively. Data duration for development and evaluation is about 3h each.

# B EVALUATION METRICS

**DNSMOS**: Deep Noise Suppression Mean Opinion Score (DNSMOS) is a non-intrusive objective metric (Reddy et al., 2022). It consists of a nerual network which was trained to predict human Mean Opinion Score (MOS) ratings for speech signal. DNSMOS will provide performance scores for the

speech signal quality (SIG), the background intrusiveness (BAK), and the overall quality (OVRL), Where each values between 1 and 5, the higher values indicating better quality. Befor computing the DNSMOS performance scores, it is reqried to use the Python package pyloudnorm to normalize the output signals at a loudness of -30 LUFS (Loudness Unit FullScale) (Steinmetz & Reiss, 2021). The motivation for this normalization is that DNSMOS scores (especially the SIG and BAK scores) are very sensitive to a change in the input signal loudness. This sensitivity would make it difficult to comparedifferent systems without a common normalization procedure.

**PESQ:** Perceptual Evaluation of Speech Quality (PESQ) (Rix et al., 2001) is a well-known intrusive speech quality measure, with a range from 1 to 4.5, making it a widely-used metric for measuring the performance of SE algorithms and the clarity of processed speech. A higher score indicates better quality.

**SI-SDR:** The Scale-Invariant Signal-to-Distortion Ratio (SI-SDR) (Le Roux et al., 2019) is a widely-used metric for assessing the quality of enhanced speech. SI-SDR quantifies the difference between the clean and estimated speech signals, measuring the improvement in signal quality while being invariant to scale changes. Higher values indicate better performance. SI-SDR is defined as follows:

$$\text{SI-SDR} = 10 \log_{10} \left( \frac{\|s_{\text{target}}\|^2}{\|e_{\text{noise}}\|^2} \right) \tag{10}$$

$$\mathbf{s}_{\text{target}} = \frac{\langle \hat{s}, s \rangle \cdot s}{\|s\|^2} \tag{11}$$

$$\mathbf{e}_{\text{noise}} = \hat{s} - s_{\text{target}} \tag{12}$$

where $s$ and $\hat{s}$ are the clean the estimated signal,respectively.

## C    VISUALIZATION

To intuitively demonstrate the effectiveness of the GGCA module, we provide sample recordings from three datasets for visualization, as shown in Fig. 6. The subfigures of each column (a) to (d) denote the spectrograms of the noisy mixture, the clean speech (or loudness-normalized mixture), the speech predicted by SS-SENet without the GGCA module, and the speech estimated by SS-SENet with GGCA, respectively. The highlighted boxes indicate regions where SS-SENet with GGCA shows noticeable improvement compared to the version without the GGCA module. Compare (d) with (c) in each subfigures, we can clear observe that the model with GGCA module exhibits better preservation of speech and more effective suppression of noise components.

## D    ABLATION STUDY

**Maximum Attenuation Rate** $\epsilon$. Since excessive attenuation ratio may lead ro the loss of information, we introduce a sclae factor $\epsilon$ that control the maximum attenuation rate in Eq. 4. As shown in Table 4, when $\epsilon$ was set to 0.1, SS-SENet achieves the best overall banlanced performance, yielding the best performance on CHiME-5 and Reverberant LibriCHiME-5 datasets, while maintaining competitive performance on LibriMix dataset. When $\epsilon$ was set to 0.05, we can observe that the overall performance is decreased on both the CHiME-5 and Reverberant LibriCHiME-5 datasets. This can be attributed to excessively conservative attenuation ceiling prevents adequate suppression of domain-specific features, thereby limiting cross-domain generalization capability. Conversely, higher values of $\epsilon$, such as 0.15 and 0.2, lead to consistent performance degradation across almost all metrics, with particularly notable declines in SI-SDR and PESQ on Reverberant LibriCHiME-5 dataset. This demonstrates that while aggressive attenuation can effectively remove domain-specific information, it simultaneously risks eliminating beneficial domain-invariant features essential for SE task. The optimal $\epsilon = 0.1$ strikes a delicate balance, providing sufficient attenuation capacity to address domain shifts while preserving critical acoustic information.

**Adjustment coefficient** $k$. To verify the effectiveness of dynamic value of $k$, we concocted ablation studies with fixed $k$ values in the DSAM strategy. As shown in Table 5, SS-SENet with dynamic adjustment coefficient $k$ achives the best performance on both CHiME-5 and Reverberant LibriCHiME-5 datasets. When $k$=10, the model achieves the highest SIG score on CHiME-5 and

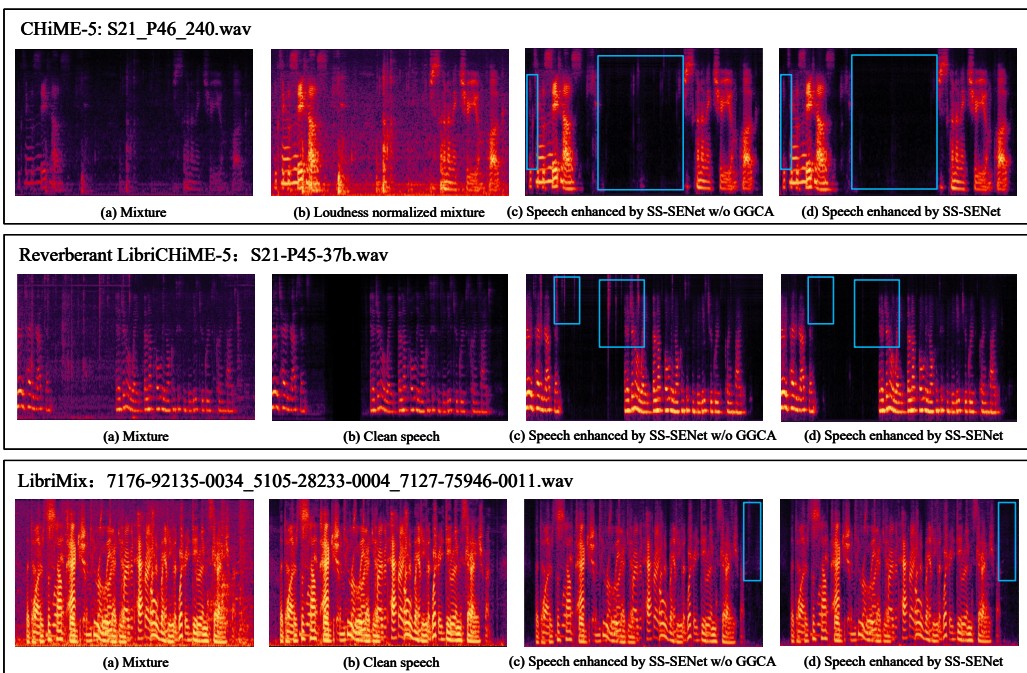

Figure 6: Visualization analysis of spectgram of three recordings, which are from three datasets respectively. The subfigures of each column from (a)–(d) denote the spectrograms of noisy mixture, clean speech (or loudness-normalized mixture for unlabeled CHiME-5 dataset), and the estimated speech by SS-SENet without the GGCA module, and the speech estimated by SS-SENet, respectively. The highlighted boxes indicate regions where SS-SENet with GGCA achieves noticeable improvement compared to the version without the GGCA module.

Table 4: Comparison of performance with different maximum attenuation ratio $\epsilon$ on three test sets.

| $\epsilon$ | CHiME-5 (Real-World) | | | Rever. LibriCHiME-5 (Synthetic) | | LibriMix (Synthetic) | |
|---|---|---|---|---|---|---|---|
| | SIG | BAK | OVRL | SI-SDR | PESQ | SI-SDR | PESQ |
| 0.05 | 3.255 | **3.856** | 2.959 | 13.255 | 1.858 | **13.086** | **2.183** |
| 0.1 | 3.338 | 3.848 | **2.971** | **13.601** | **1.894** | 13.043 | 2.164 |
| 0.15 | **3.354** | 3.775 | 2.969 | 12.443 | 1.685 | 12.992 | 2.139 |
| 0.2 | 3.299 | 3.813 | 2.965 | 12.481 | 1.671 | 12.975 | 2.138 |

the best performance on the LibriMix dataset, but suffers from a significant degradation in SI-SDR on Reverberant LibriCHiME-5 dataset. Similarly, $k=15$ produces the second-best SI-SDR on Reverberant LibriCHiME-5 but fails to maintain high OVRL scores on CHiME-5 dataset. Notably, both overly conservative ($k=5$) and overly aggressive ($k=20$) settings result in suboptimal performance on almost all metrics. These reults verify that a fixed slope values cannot adapt to the varying domain-shift conditions. The dynamic adjustment of $k$ based on $\nu$ enables Eq. 4 adaptively modulate its sensitivity, achieving robust and balanced performance during training process.

**Consistency loss weight** $\lambda 1$. To further analyze the importance of consistency constraints in our proposed SS-SENet, we conducted ablation studies on the weight $\lambda_1$ of the consistency loss $\mathcal{L}_{\text{Cons.}}$, while fixing $\lambda 2$ of the domain loss $\mathcal{L}_{\text{Dom.}}$ at 0.1. The results are listed in Table 6. When $\lambda 1$ is set to 1.0, it can be observed that SS-SENet achieves nearly the best performance on both the CHiME-5 and Reverberant LibriCHiME-5 datasets, with particularly notable improvements in the item of OVRL on CHiME-5. When $\lambda_1$ is reduced to smaller values such as 0.3 or 0.5, the performance of SS-SENet clearly degrades on both CHiME-5 and Reverberant LibriCHiME-5. This indicates that

Table 5: Result of ablation experiments of dynamic coefficient $k$ in DSAM on three test sets.

| Method | CHiME-5 (Real-World) | | | Rever. LibriCHiME-5 (Synthetic) | | LibriMix (Synthetic) | |
|---|---|---|---|---|---|---|---|
| | SIG | BAK | OVRL | SI-SDR | PESQ | SI-SDR | PESQ |
| SS-SENet | 3.338 | **3.848** | **2.971** | **13.601** | **1.894** | 13.043 | 2.164 |
| k=5 | 3.362 | 3.726 | 2.936 | 12.943 | 1.721 | 13.042 | 2.168 |
| k=10 | **3.400** | 3.711 | 2.954 | 11.015 | 1.748 | **13.091** | **2.185** |
| k=15 | 3.371 | 3.792 | 2.950 | 13.429 | 1.868 | 13.050 | 2.156 |
| k=20 | 3.312 | 3.803 | 2.948 | 13.155 | 1.856 | 13.046 | 2.151 |

Table 6: Result of ablation experiments of consistency loss weight $\lambda 1$ on three test sets.

| $\lambda 1$ | CHiME-5 (Real-World) | | | Rever. LibriCHiME-5 (Synthetic) | | LibriMix (Synthetic) | |
|---|---|---|---|---|---|---|---|
| | SIG | BAK | OVRL | SI-SDR | PESQ | SI-SDR | PESQ |
| 0.3 | **3.348** | 3.633 | 2.947 | 11.564 | 1.642 | 13.186 | 2.174 |
| 0.5 | 3.289 | 3.834 | 2.948 | 13.547 | 1.864 | **13.236** | **2.221** |
| 1 | 3.338 | **3.848** | **2.971** | **13.601** | **1.894** | 13.043 | 2.164 |
| 2 | 3.228 | 3.302 | 2.756 | 8.724 | 1.565 | 11.236 | 1.775 |

insufficient consistency regularization weakens the adaptation ability. In contrast, when $\lambda_1$ is further increased, such as 2.0, severe performance degradation is observed across all datasets. This suggests that an excessively large $\lambda 1$ causes the consistency training to dominate training, thereby hindering optimization of the primary enhancement objective.