# OpenReview forum: "Semi-Supervised Speech Enhancement with Gradient-Guided Channel Attenuation"
_ICLR.cc/2026/Conference — ICLR 2026 Conference Withdrawn Submission_

### Official Review · Reviewer_yN4M · 2025-10-21

**Soundness:** 2
**Presentation:** 2
**Contribution:** 1
**Rating:** 2
**Confidence:** 3

**Summary:**

The paper proposes SS-SENet, a semi-supervised monaural speech enhancement network that integrates a Mean Teacher (MT) framework with domain-adversarial (DA) learning to leverage unlabeled real-world audio. It introduces two modules: Gradient-Guided Channel Attenuation (GGCA), which suppresses domain-specific features, and Domain Shift-Aware Monitor (DSAM), which dynamically adjusts attenuation strength. Trained on LibriMix and CHiME-5 datasets, SS-SENet outperforms seven state-of-the-art baselines on synthetic and real-world benchmarks, demonstrating improved generalization and robustness.

**Strengths:**

1. Comprehensive Experimental Evaluation: The paper presents extensive experimental results on both synthetic and real-world datasets (CHiME-5, LibriMix, and Reverberant LibriCHiME-5). The comparisons include seven competing baselines, covering both self-supervised and domain-adaptive approaches, which strengthens the empirical validation.

2. Transparent Code Availability: The authors provide open-source code, which facilitates reproducibility and increases the paper’s practical value for future research in semi-supervised speech enhancement.

3. Thorough Ablation Studies: The paper conducts systematic ablation analyses on each major component (DA, GGCA, and DSAM), clearly demonstrating their individual contributions to overall performance improvements.

**Weaknesses:**

1. Limited Contribution and Novelty: The proposed framework primarily combines existing ideas such as Mean Teacher consistency learning and domain-adversarial training without introducing a fundamentally novel algorithmic or theoretical component. As a result, the methodological contribution feels incremental rather than innovative.

2. Insufficient Discussion of Noise and Speech Types: The paper could benefit from a more detailed characterization of the noise types, speech conditions, and specific challenges encountered in the real-world CHiME-5 dataset. Such clarification would better contextualize the proposed method’s robustness and its potential limitations in practical deployment.

3. Language and Presentation Issues: The manuscript contains several typographical and grammatical errors (e.g., “expoit”, “extratced”, “sturcture”, “banlanced”, “adpot”) throughout Section 3 and the ablation appendices. Although these do not hinder overall comprehension, they detract from the paper’s professionalism and polish.

4. Lack of Statistical Significance Analysis: The reported improvements are not accompanied by statistical significance tests (e.g., Wilcoxon signed-rank test or paired t-test). This omission makes it difficult to assess whether the observed gains are consistent or potentially due to random variation.

5. Absence of Efficiency Metrics: The paper does not provide any computational efficiency measurements, such as training speed, parameter count, GPU memory usage, or FLOPs. This omission limits the reader’s ability to assess the trade-off between performance gains and computational cost.

**Questions:**

1. How consistent are the reported improvements across different random seeds or training runs? Have the authors performed any statistical tests to verify that the performance gains are significant?

2. Given that the Mean Teacher framework doubles the number of model instances during training, what is the additional computational cost in terms of runtime, training speed, parameter count, GPU memory usage, and FLOPs compared to purely supervised baselines?

3. Could the authors provide more detailed information on the types and distributions of noise in the real-world CHiME-5 dataset, and whether specific noise characteristics affect model performance?

4. The study focuses primarily on SI-SDR, PESQ, and DNSMOS metrics. Have the authors considered perceptual or subjective listening tests to validate whether the improvements are perceptually meaningful?

---

### Official Review · Reviewer_3bsn · 2025-10-23

**Soundness:** 2
**Presentation:** 3
**Contribution:** 2
**Rating:** 2
**Confidence:** 5

**Summary:**

This paper proposes a semi-supervised monaural speech enhancement network, SSSENet, which adopts the Mean-Teacher (MT) framework with domain adversarial (DA) learning. It also introduces a Gradient-Guided Channel Attenuation (GGCA) module to suppress domain-specific features and enhance domain-invariant ones, along with a Domain Shift-Aware Monitor (DSAM) strategy to dynamically adjust the attenuation rate in GGCA, enabling more effective use of unlabeled data. However, there are some concerns regarding the experimental setup and the limited performance improvements observed (see the weakness below).

**Strengths:**

1)	The ideas of GGCA and DSAM are interesting, although they seem to serve the same purpose as domain adversarial (DA) learning.
2)	The paper is well written and easy to follow.

**Weaknesses:**

1)	Since the domain discrimination loss has already been applied to remove domain-specific information, the necessity of the proposed Gradient-Guided Channel Attenuation (GGCA) and Domain Shift-Aware Monitor (DSAM) is unclear. In fact, as shown in Table 3, the improvements over domain adversarial (DA) training are quite limited and could be within the range of normal fluctuations due to different training iterations. I’m curious whether, if a learning curve plot similar to Figure 5 were shown, we could clearly observe any improvement from GGCA and DSAM.
2)	In Table 6 (Appendix), the authors present ablation studies on hyperparameter tuning and report results on the test set. This approach may lead to overfitting and is not a fair evaluation, as test data should not be used for parameter tuning.

**Questions:**

1)	For the seven baseline models presented in the experiments, were they all trained on the same dataset as your model?
2)	In Figure 5, could you explain why RemixIT does not show improved scores?
3)	In Table 2, why does Basic MT significantly outperform SuDoRM-RF (F) on the Reverb. LibriCHiME-5 dataset but not on CHiME-5? Since MT is semi-supervised on CHiME-5, I’m quite curious about the reason behind this discrepancy.

---

### Official Review · Reviewer_VdMs · 2025-10-30

**Soundness:** 2
**Presentation:** 3
**Contribution:** 3
**Rating:** 6
**Confidence:** 3

**Summary:**

The paper proposes a novel semi-supervised approach for speech enhancement that enables the incorporation of real-life data without requiring direct reference to clean samples in the training pipeline. The paper also introduces techniques to enhance the efficacy of semi-supervised training; in particular, it proposes attenuating domain-specific features to improve the model's generalisation to other domains and utilises a domain adversarial loss to make features less susceptible to domain changes.

**Strengths:**

The paper has the following strengths:

1. The idea of using semi-supervised methods to incorporate real-world data without clean reference recordings has significant potential, as it allows models to be trained on a much more diverse set of recordings.

2. The structure of the paper is good; it is well-structured and easy to read. Both the model architecture and the training pipeline are well-detailed, making it easy to comprehend the key components of the proposed algorithm.

**Weaknesses:**

The paper has some minor weaknesses:

1. The paper lacks comparisons with recent universal SE models trained on paired data. Since the paper claims that incorporating unlabelled data is beneficial, it would be interesting to see how the method compares to other recent models [1, 2, 3, 4, 5, 6] that use s traditional approach.

2. The DNSMOS results seem strange. They show that the model degrades the perceptual quality of the audio. I would recommend adding UTMOS [7]; moreover,  providing some reference audio recordings would be beneficial for a better understanding of the performance of the model.

**Questions:**

1. Can the proposed semi-supervised method be used together with existing GAN-based or diffusion-based pipelines, which are widely adopted in a large body of work?

#### **References**

[1] Babaev et al., "FINALLY: fast and universal speech enhancement with studio-like quality".

[2] Su et al., "HiFi-GAN-2: studio-quality speech enhancement via generative".

[3] Lemercier et al., "StoRM: a diffusion-based stochastic regeneration model for speech enhancement and dereverberation".

[4] Scheibler et al.,  "Universal score-based speech enhancement with high content preservation".

[5] Jukíc et al., "Schrödinger bridge for generative speech enhancement".

[6] Wang et al., "Diffusion-based Speech Enhancement with Schrödinger Bridge and Symmetric Noise Schedule".

[7] Saeki et al., "UTMOS: UTokyo-SaruLab system for VoiceMOS challenge 2022"

---

### Official Review · Reviewer_Lesi · 2025-11-01

**Soundness:** 3
**Presentation:** 3
**Contribution:** 2
**Rating:** 4
**Confidence:** 4

**Summary:**

The paper considers the speech enhancement problem. The authors propose a semi-supervised monaural speech enhancement method based on the mean-teacher approach. A domain discriminator with gradient reversal layer is designed to narrow the synthetic-real gap. A gradient-guided channel attenuation module is emplyed to use discriminator gradients to rank channel-wise domain specificity and attenuate Top-k channels, while domain shift aware monitor is used to adapt the attenuation ratio via a batch-level labeled/unlabeled feature variance measure. Provided experiments on CHiME-5 and LibriMix/Reverberant LibriCHiME-5 show the proposed method outperforms recent methods across backbones.

**Strengths:**

The paper applies the mean teacher method to speech enhancement, and combines labeled synthetic data and unlabeled real-world data in a single-stage speech enhancement training pipeline, which is simpler than two-stage methods like RemixIT.  A gradient-guided channel attenuation (GGCA) module for for suppressing domain-specific features and enhancing domain-invariant features, and a domain shift aware monitor strategy for dynamically controlling the attenuation rate in GGCA have been proposed to further improve the performacne. Experimental results are reported across two backbone types, multiple datasets, seven baseline methods, which demonstrate the effectiveness of the proposed method.

**Weaknesses:**

The contribution of this paper lies mainly in combining existing methods, mean teacher, domain adversarial learning, channel suppression, into a system for speech enhancement. The novelty is more in engineering integration of existing methods. It would be better to include downstream task results such as ASR word error rate, to validate practical usefulness of the proposed method.

**Questions:**

Does mean teacher remain stable when unlabeled data dominates? How does it handle pseudo-label noise accumulation?
It would be better to include downstream task results such as ASR WER.

---

### Note · Authors · 2025-11-17

I have read and agree with the venue's withdrawal policy on behalf of myself and my co-authors.